# DialogWAE: Multimodal Response Generation with Conditional Wasserstein Auto-Encoder

**Xiaodong Gu**[1,3]**, Kyunghyun Cho**[2,4]**, Jung-Woo Ha**[3]**, Sunghun Kim**[1,3]
[1]Hong Kong University of Science and Technology,
[2]New York Universidy, [3]Clova AI Research, NAVER, [4]CIFAR Azrieli Global Scholar
[1]`guxiaodong1987@126.com, hunkim@cse.ust.hk`
[2]`kyunghyun.cho@nyu.edu,` [3]`jungwoo.ha@navercorp.com`

## Abstract

Variational autoencoders (VAEs) have shown a promise in data-driven conversation modeling. However, most VAE conversation models match the approximate posterior distribution over the latent variables to a simple prior such as standard normal distribution, thereby restricting the generated responses to a relatively simple (e.g., unimodal) scope. In this paper, we propose DialogWAE, a conditional Wasserstein autoencoder (WAE) specially designed for dialogue modeling. Unlike VAEs that impose a simple distribution over the latent variables, DialogWAE models the distribution of data by training a GAN within the latent variable space. Specifically, our model samples from the prior and posterior distributions over the latent variables by transforming context-dependent random noise using neural networks and minimizes the Wasserstein distance between the two distributions. We further develop a Gaussian mixture prior network to enrich the latent space. Experiments on two popular datasets show that DialogWAE outperforms the state-of-the-art approaches in generating more coherent, informative and diverse responses.

## 1 Introduction

Neural response generation has been a long interest of natural language research. Most of the recent approaches to data-driven conversation modeling primarily build upon sequence-to-sequence learning (Cho *et al.*, 2014; Sutskever *et al.*, 2014). Previous research has demonstrated that sequence-to-sequence conversation models often suffer from the *safe response* problem and fail to generate meaningful, diverse on-topic responses (Li *et al.*, 2015; Sato *et al.*, 2017). Conditional variational autoencoders (CVAE) have shown promising results in addressing the safe response issue (Zhao *et al.*, 2017; Shen *et al.*, 2018). CVAE generates the response conditioned on a latent variable - representing topics, tones and situations of the response - and approximate the posterior distribution over latent variables using a neural network. The latent variable captures variabilities in the dialogue and thus generates more diverse responses. However, previous studies have shown that VAE models tend to suffer from the posterior collapse problem, where the decoder learns to ignore the latent variable and degrades to a vanilla RNN (Shen *et al.*, 2018; Park *et al.*, 2018; Bowman *et al.*, 2015). Furthermore, they match the approximate posterior distribution over the latent variables to a simple prior such as standard normal distribution, thereby restricting the generated responses to a relatively simple (e.g., unimodal) scope (Goyal *et al.*, 2017).

A number of studies have sought GAN-based approaches (Goodfellow *et al.*, 2014; Li *et al.*, 2017a; Xu *et al.*, 2017) which directly model the distribution of the responses. However, adversarial training over discrete tokens has been known to be difficult due to the non-differentiability. Li *et al.* (2017a) proposed a hybrid model of GAN and reinforcement learning (RL) where the score predicted by a discriminator is used as a reward to train the generator. However, training with REINFORCE has been observed to be unstable due to the high variance of the gradient estimate (Shen *et al.*, 2017). Xu *et al.* (2017) make the GAN model differentiable with an approximate word embedding layer. However, their model only injects variability at the word level, thus limited to represent high-level response variabilities such as topics and situations.

In this paper, we propose DialogWAE, a novel variant of GAN for neural conversation modeling. Unlike VAE conversation models that impose a simple distribution over latent variables, DialogWAE models the data distribution by training a GAN within the latent variable space. Specifically, it samples from the prior and posterior distributions over the latent variables by transforming context-dependent random noise with neural networks, and minimizes the Wasserstein distance (Arjovsky *et al.*, 2017) between the prior and the approximate posterior distributions. Furthermore, our model takes into account a multimodal[1] nature of responses by using a Gaussian mixture prior network. Adversarial training with the Gaussian mixture prior network enables DialogWAE to capture a richer latent space, yielding more coherent, informative and diverse responses.

Our main contributions are two-fold: (1) A novel GAN-based model for neural dialogue modeling, which employs GAN to generate samples of latent variables. (2) A Gaussian mixture prior network to sample random noise from a multimodal prior distribution. To the best of our knowledge, the proposed DialogWAE is the first GAN conversation model that exploits multimodal latent structures.

We evaluate our model on two benchmark datasets, SwitchBoard (Godfrey and Holliman, 1997) and DailyDialog (Li *et al.*, 2017b). The results demonstrate that our model substantially outperforms the state-of-the-art methods in terms of BLEU, word embedding similarity, and distinct. Furthermore, we highlight how the GAN architecture with a Gaussian mixture prior network facilitates the generation of more diverse and informative responses.

## 2 RELATED WORK

**Encoder-decoder variants** To address the "safe response" problem of the naive encoder-decoder conversation model, a number of variants have been proposed. Li *et al.* (2015) proposed a diversity-promoting objective function to encourage more various responses. Sato *et al.* (2017) propose to incorporate various types of situations behind conversations when encoding utterances and decoding their responses, respectively. Xing *et al.* (2017) incorporate topic information into the sequence-to-sequence framework to generate informative and interesting responses. Our work is different from the aforementioned studies, as it does not rely on extra information such as situations and topics.

**VAE conversation models** The variational autoencoder (VAE) (Kingma and Welling, 2014) is among the most popular frameworks for dialogue modeling (Zhao *et al.*, 2017; Shen *et al.*, 2018; Park *et al.*, 2018). Serban *et al.* (2017) propose VHRED, a hierarchical latent variable sequence-to-sequence model that explicitly models multiple levels of variability in the responses. A main challenge for the VAE conversation models is the so-called "posterior collapse". To alleviate the problem, Zhao *et al.* (2017) introduce an auxiliary bag-of-words loss to the decoder. They further incorporate extra dialogue information such as dialogue acts and speaker profiles. Shen *et al.* (2018) propose a collaborative CVAE model which samples the latent variable by transforming a Gaussian noise using neural networks and matches the prior and posterior distributions of the Gaussian noise with KL divergence. Park *et al.* (2018) propose a variational hierarchical conversation RNN (VHCR) which incorporates a hierarchical structure to latent variables. DialogWAE addresses the limitation of VAE conversation models by using a GAN architecture in the latent space.

**GAN conversation models** Although GAN/CGAN has shown great success in image generation, adapting it to natural dialog generators is a non-trivial task. This is due to the non-differentiable nature of natural language tokens (Shen *et al.*, 2017; Xu *et al.*, 2017). Li *et al.* (2017a) address this problem by combining GAN with Reinforcement Learning (RL) where the discriminator predicts a reward to optimize the generator. However, training with REINFORCE can be unstable due to the high variance of the sampled gradient (Shen *et al.*, 2017). Xu *et al.* (2017) make the sequence-to-sequence GAN differentiable by directly multiplying the word probabilities obtained from the decoder to the corresponding word vectors, yielding an approximately vectorized representation of the target sequence. However, their approach injects diversity in the word level rather than the level of the whole responses. DialogWAE differs from exiting GAN conversation models in that it shapes the distribution of responses in a high level latent space rather than direct tokens and does not rely on RL where the gradient variances are large.

---

[1]A multimodal distribution is a continuous probability distribution with two or more modes.

## 3 PROPOSED APPROACH

### 3.1 PROBLEM STATEMENT

Let $d=[u_1, ..., u_k]$ denote a dialogue of $k$ utterances where $u_i=[w_1, ..., w_{|u_i|}]$ represents an utterance and $w_n$ denotes the $n$-th word in $u_i$. Let $c=[u_1, ..., u_{k-1}]$ denote a dialogue context, the $k$-1 historical utterances, and $x=u_k$ be a response which means the next utterance. Our goal is to estimate the conditional distribution $p_\theta(x|c)$.

As $x$ and $c$ are sequences of discrete tokens, it is non-trivial to find a direct coupling between them. Instead, we introduce a continuous latent variable $z$ that represents the high-level representation of the response. The response generation can be viewed as a two-step procedure, where a latent variable $z$ is sampled from a distribution $p_\theta(z|c)$ on a latent space $\mathcal{Z}$, and then the response $x$ is decoded from $z$ with $p_\theta(x|z, c)$. Under this model, the likelihood of a response is

$$p_\theta(x|c) = \int_z p(x|c, z)p(z|c)d_z. \tag{1}$$

The exact log-probability is difficult to compute since it is intractable to marginalize out $z$. Therefore, we approximate the posterior distribution of $z$ as $q_\phi(z|x, c)$ which can be computed by a neural network named *recognition network*. Using this approximate posterior, we can instead compute the evidence lower bound (ELBO):

$$\begin{aligned} \log p_\theta(x|c) &= \log \int_z p(x|c, z)p(z|c)d_z \\ &\geq \ell(x, c) = \mathbf{E}_{z \sim q_\phi(z|x,c)}[\log p_\psi(x|c, z)] - \mathrm{KL}(q_\phi(z|x, c)||p(z|c)), \end{aligned} \tag{2}$$

where $p(z|c)$ represents the prior distribution of $z$ given $c$ and can be modeled with a neural network named *prior network*.

### 3.2 CONDITIONAL WASSERSTEIN AUTO-ENCODERS FOR DIALOGUE MODELING

The conventional VAE conversation models assume that the latent variable $z$ follows a simple prior distribution such as the normal distribution. However, the latent space of real responses is more complicated and difficult to be estimated with such a simple distribution. This often leads to the posterior collapse problem (Shen *et al.*, 2018).

Inspired by GAN and the adversarial auto-encoder (AAE) (Makhzani *et al.*, 2015; Tolstikhin *et al.*, 2017; Zhao *et al.*, 2018), we model the distribution of $z$ by training a GAN within the latent space. We sample from the prior and posterior over the latent variables by transforming random noise $\epsilon$ using neural networks. Specifically, the prior sample $\tilde{z} \sim p_\theta(z|c)$ is generated by a generator $G$ from context-dependent random noise $\tilde{\epsilon}$, while the approximate posterior sample $z \sim q_\phi(z|c, x)$ is generated by a generator $Q$ from context-dependent random noise $\epsilon$. Both $\tilde{\epsilon}$ and $\epsilon$ are drawn from a normal distribution whose mean and covariance matrix (assumed diagonal) are computed from $c$ with feed-forward neural networks, *prior network* and *recognition network*, respectively:

$$\tilde{z} = G_\theta(\tilde{\epsilon}), \quad \tilde{\epsilon} \sim \mathcal{N}(\epsilon; \tilde{\mu}, \tilde{\sigma}^2 I), \quad \begin{bmatrix} \tilde{\mu} \\ \log \tilde{\sigma}^2 \end{bmatrix} = \tilde{W}f_\theta(c) + \tilde{b} \tag{3}$$

$$z = Q_\phi(\epsilon), \quad \epsilon \sim \mathcal{N}(\epsilon; \mu, \sigma^2 I), \quad \begin{bmatrix} \mu \\ \log \sigma^2 \end{bmatrix} = Wg_\phi(\begin{bmatrix} x \\ c \end{bmatrix}) + b, \tag{4}$$

where $f_\theta(\cdot)$ and $g_\phi(\cdot)$ are feed-forward neural networks. Our goal is to minimize the divergence between $p_\theta(z|c)$ and $q_\phi(z|x, c)$ while maximizing the log-probability of a reconstructed response from $z$. We thus solve the following problem:

$$\min_{\theta, \phi, \psi} -E_{q_\phi(z|x,c)} \log p_\psi(x|z, c) + W(q_\phi(z|x, c)||p_\theta(z|c)), \tag{5}$$

where $p_\theta(z|c)$ and $q_\phi(z|x, c)$ are neural networks implementing Equations 3 and 4, respectively. $p_\psi(x|z, c)$ is a decoder. $\mathrm{W}(\cdot||\cdot)$ represents the Wasserstein distance between these two distributions (Arjovsky *et al.*, 2017). We choose the Wasserstein distance as the divergence since the WGAN has been shown to produce good results in text generation (Zhao *et al.*, 2018).

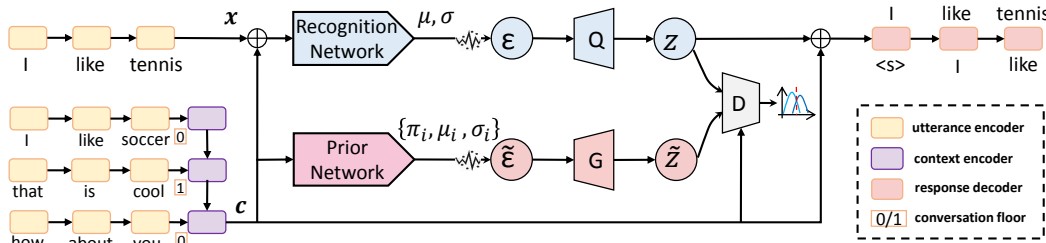

Figure 1: Architecture of DialogWAE

Figure 1 illustrates an overview of our model. The *utterance encoder* (RNN) transforms each utterance (including the response $x$) in the dialogue into a real-valued vector. For the $i$-th utterance in the context, the *context encoder* (RNN) takes as input the concatenation of its encoding vector and the conversation floor (1 if the utterance is from the speaker of the response, otherwise 0) and computes its hidden state $h_i^{ctx}$. The final hidden state of the context encoder is used as the context representation.

At generation time, the model draws a random noise $\tilde{\epsilon}$ from the *prior network* (PriNet) which transforms $c$ through a feed-forward network followed by two matrix multiplications which result in the mean and diagonal covariance, respectively. Then, the generator G generates a sample of latent variable $\tilde{z}$ from the noise through a feed-forward network. The *decoder* RNN decodes the generated $\tilde{z}$ into a response.

At training time, the model infers the posterior distribution of the latent variable conditioned on the context $c$ and the response $x$. The *recognition network* (RecNet) takes as input the concatenation of both $x$ and $c$ and transforms them through a feed-forward network followed by two matrix multiplications which define the normal mean and diagonal covariance, respectively. A Gaussian noise $\epsilon$ is drawn from the recognition network with the re-parametrization trick. Then, the *generator* Q transforms the Gaussian noise $\epsilon$ into a sample of latent variable $z$ through a feed-forward network. The response *decoder* (RNN) computes the reconstruction loss:

$$\mathcal{L}_{rec} = -E_{z=Q(\epsilon), \ \epsilon \sim \text{RecNet}(x,c)} \log p_\psi(x|c, z) \tag{6}$$

We match the approximate posterior with the prior distributions of $z$ by introducing an adversarial discriminator D which tells apart the prior samples from posterior samples. D is implemented as a feed-forward neural network which takes as input the concatenation of $z$ and $c$ and outputs a real value. We train D by minimizing the discriminator loss:

$$\mathcal{L}_{disc} = E_{\epsilon \sim \text{RecNet}(x,c)}[D(Q(\epsilon), c)] - E_{\tilde{\epsilon} \sim \text{PriNet}(c)}[D(G(\tilde{\epsilon}), c)] \tag{7}$$

### 3.3 MULTIMODAL RESPONSE GENERATION WITH A GAUSSIAN MIXTURE PRIOR NETWORK

It is a usual practice for the prior distribution in the AAE architecture to be a normal distribution. However, responses often have a multimodal nature reflecting many equally possible situations (Sato *et al.*, 2017), topics and sentiments. A random noise with normal distribution could restrict the generator to output a latent space with a single dominant mode due to the unimodal nature of Gaussian distribution. Consequently, the generated responses could follow simple prototypes.

To capture multiple modes in the probability distribution over the latent variable, we further propose to use a distribution that explicitly defines more than one mode. Each time, the noise to generate the latent variable is selected from one of the modes. To achieve so, we make the prior network to capture a mixture of Gaussian distributions, namely, $\text{GMM}(\{\pi_k, \mu_k, \sigma_k^2 I\}_{k=1}^K)$, where $\pi_k$, $\mu_k$ and $\sigma_k$ are parameters of the $k$-th component. This allows it to learn a multimodal manifold in the latent variable space in a two-step generation process – first choosing a component $k$ with $\pi_k$, and then sampling Gaussian noise within the selected component:

$$p(\epsilon|c) = \sum_{k=1}^{K} v_k \mathcal{N}(\epsilon; \mu_k, \sigma_k^2 I), \tag{8}$$

---

**Algorithm 1:** DialogWAE Training (UEnc: utterance encoder; CEnc: context encoder; RecNet: recognition network; PriNet: prior network; Dec: decoder) K=3, $n_{\text{critic}}$=5 in all experiments

---

**In:** a dialog corpus $\mathcal{D}$={$(c_i, x_i)$}$_{i=1}^{|\mathcal{D}|}$, the number of prior modes $K$, discriminator iterations $n_{\text{critic}}$

1  Initialize {$\theta_{\text{UEnc}}, \theta_{\text{CEnc}}, \theta_{\text{PriNet}}, \theta_{\text{RecNet}}, \theta_Q, \theta_G, \theta_D, \theta_{\text{Dec}}$}

2  **while** *not convergence* **do**

3  |  Initialize $\mathcal{D}$

4  |  **while** $\mathcal{D}$ has unsampled batches **do**

5  |  |  Sample a mini-batch of N instances {$(x_n, c_n)$}$_{n=1}^N$ from $\mathcal{D}$

6  |  |  Get the representations of context and response $\boldsymbol{x}_n$=UEnc($x_n$), $\boldsymbol{c}_n$=CEnc($c_n$)

7  |  |  Sample $\boldsymbol{\epsilon}_n$ from RecNet($\boldsymbol{x}_n, \boldsymbol{c}_n$) according to Equation 4

8  |  |  Sample $\hat{\boldsymbol{\epsilon}}_n$ from PriNet($\boldsymbol{c}_n, K$) according to Equation 8–10

9  |  |  Generate $z_n$ = Q($\boldsymbol{\epsilon}_n$), $\tilde{z}_n$ = G($\hat{\boldsymbol{\epsilon}}_n$)

10 |  |  Update {$\theta_Q, \theta_G, \theta_{\text{PriNet}}, \theta_{\text{RecNet}}$} by gradient ascent on discriminator loss

11 |  |  $\quad\quad \mathcal{L}_{disc} = \frac{1}{N}\sum_{n=1}^N D(z_n, c_n) - \frac{1}{N}\sum_{n=1}^N D(\tilde{z}_n, c_n)$

12 |  |  **for** $i \in \{1, \cdots, n_{\text{critic}}\}$ **do**

13 |  |  |  Repeat 5–9

14 |  |  |  Update $\theta_D$ by gradient descent on the discriminator loss $\mathcal{L}_{disc}$ with gradient penalty

15 |  |  **end**

16 |  |  Update {$\theta_{\text{UEnc}}, \theta_{\text{CEnc}}, \theta_{\text{RecNet}}, \theta_Q, \theta_{\text{Dec}}$} by gradient descent on the reconstruction loss

17 |  |  $\quad\quad \mathcal{L}_{rec} = -\frac{1}{N}\sum_{n=1}^N \log p(x_n|z_n, c_n)$

18 |  **end**

19 **end**

---

where $v_k \in \Delta^{K-1}$ is a component indicator with class probabilities $\pi_1, \cdots, \pi_K$; $\pi_k$ is the mixture coefficient of the $k$-th component of the GMM. They are computed as

$$\boldsymbol{\pi}_k = \frac{\exp(e_k)}{\sum_{i=1}^K \exp(e_i)}, \quad \text{where} \quad \begin{bmatrix} e_k \\ \mu_k \\ \log \sigma_k^2 \end{bmatrix} = W_k f_\theta(c) + b_k \tag{9}$$

Instead of exact sampling, we use Gumbel-Softmax re-parametrization (Kusner and Hernández-Lobato, 2016) to sample an instance of $v$:

$$v_k = \frac{\exp((e_k + g_k)/\tau)}{\sum_{i=1}^K \exp((e_i + g_i)/\tau)}, \tag{10}$$

where $g_i$ is a Gumbel noise computed as

$$g_i = -\log(-\log(u_i)), u_i \sim U(0, 1)$$

and $\tau \in [0,1]$ is the softmax temperature which is set to 0.1 in all experiments.

We refer to this framework as DialogWAE-GMP. A comparison of performance with different numbers of prior components will be shown in Section 5.1.

### 3.4 TRAINING

Our model is trained epochwise until a convergence is reached. In each epoch, we train the model iteratively by alternating two phases − an AE phase during which the reconstruction loss of decoded responses is minimized, and a GAN phase which minimizes the Wasserstein distance between the prior and approximate posterior distributions over the latent variables. The detailed procedures are presented in Algorithm 1

## 4 EXPERIMENTAL SETUP

**Datasets** We evaluate our model on two dialogue datasets, Dailydialog (Li *et al.*, 2017b) and Switchboard (Godfrey and Holliman, 1997), which have been widely used in recent studies (Shen *et al.*, 2018; Zhao *et al.*, 2017). Dailydialog has 13,118 daily conversations for a English learner in a

daily life. Switchboard contains 2,400 two-way telephone conversations under 70 specified topics. The datasets are separated into training, validation, and test sets with the same ratios as in the baseline papers, that is, 2316:60:62 for Switchboard (Zhao *et al.*, 2017) and 10:1:1 for Dailydialog (Shen *et al.*, 2018), respectively.

**Metrics** To measure the performance of DialogWAE, we adopted several standard metrics widely used in existing studies: BLEU (Papineni *et al.*, 2002), BOW Embedding (Liu *et al.*, 2016) and distinct (Li *et al.*, 2015). In particular, BLEU measures how much a generated response contains $n$-gram overlaps with the reference. We compute BLEU scores for n<4 using smoothing techniques (smoothing 7)[2] (Chen and Cherry, 2014). For each test context, we sample 10 responses from the models and compute their BLEU scores. We define $n$-gram precision and $n$-gram recall as the average and the maximum score respectively (Zhao *et al.*, 2017).

BOW embedding metric is the cosine similarity of bag-of-words embeddings between the hypothesis and the reference. We use three metrics to compute the word embedding similarity: 1. **Greedy**: greedily matching words in two utterances based on the cosine similarities between their embeddings, and to average the obtained scores (Rus and Lintean, 2012). 2. **Average**: cosine similarity between the averaged word embeddings in the two utterances (Mitchell and Lapata, 2008). 3. **Extrema**: cosine similarity between the largest extreme values among the word embeddings in the two utterances (Forgues *et al.*, 2014). We use Glove vectors (Pennington *et al.*, 2014) as the embeddings which will be discussed later in this section. For each test context, we report the maximum BOW embedding score among the 10 sampled responses.

*Distinct* computes the diversity of the generated responses. *dist-n* is defined as the ratio of unique $n$-grams (n=1,2) over all $n$-grams in the generated responses. As we sample multiple responses for each test context, we evaluate diversities for both within and among the sampled responses. We define *intra-dist* as the average of distinct values within each sampled response and *inter-dist* as the distinct value among all sampled responses.

**Baselines** We compare the performance of DialogWAE with seven recently-proposed baselines for dialogue modeling: (i) HRED: a generalized sequence-to-sequence model with hierarchical RNN encoder (Serban *et al.*, 2016), (ii) SeqGAN: a GAN based model for sequence generation (Li *et al.*, 2017a), (iii) CVAE: a conditional VAE model with KL-annealing (Zhao *et al.*, 2017), (iv) CVAE-BOW: a conditional VAE model with a BOW loss (Zhao *et al.*, 2017), (v) CVAE-CO: a collaborative conditional VAE model (Shen *et al.*, 2018), (vi) VHRED: a hierarchical VAE model (Serban *et al.*, 2017), and (vii) VHCR: a hierarchical VAE model with conversation modeling (Park *et al.*, 2018).

**Training and Evaluation Details** We use the gated recurrent units (GRU) (Cho *et al.*, 2014) for the RNN encoders and decoders. The utterance encoder is a bidirectional GRU with 300 hidden units in each direction. The context encoder and decoder are both GRUs with 300 hidden units. The prior and the recognition networks are both 2-layer feed-forward networks of size 200 with tanh non-linearity. The generators $Q$ and $G$ as well as the discriminator $D$ are 3-layer feed-forward networks with ReLU non-linearity (Nair and Hinton, 2010) and hidden sizes of 200, 200 and 400, respectively. The dimension of a latent variable $z$ is set to 200. The initial weights for all fully connected layers are sampled from a uniform distribution [-0.02, 0.02]. The gradient penalty is used when training $D$ (Gulrajani *et al.*, 2017) and its hyper-parameter $\lambda$ is set to 10. We set the vocabulary size to 10,000 and define all the out-of-vocabulary words to a special token <unk>. The word embedding size is 200 and initialized with Glove vectors pre-trained on Twitter (Pennington *et al.*, 2014). The size of context window is set to 10 with a maximum utterance length of 40. We sample responses with greedy decoding so that the randomness entirely come from the latent variables. The baselines were implemented with the same set of hyper-parameters. All the models are implemented with Pytorch 0.4.0[3], and fine-tuned with NAVER Smart Machine Learning (NSML) platform (Sung *et al.*, 2017; Kim *et al.*, 2018).

The models are trained with mini-batches containing 32 examples each in an end-to-end manner. In the AE phase, the models are trained by SGD with an initial learning rate of 1.0 and gradient clipping at 1 (Pascanu *et al.*, 2013). We decay the learning rate by 40% every 10th epoch. In the GAN phase, the models are updated using RMSprop (Tieleman and Hinton) with fixed learning rates of $5 \times 10^{-5}$

---

[2]`https://www.nltk.org/_modules/nltk/translate/bleu_score.html`
[3]`https://pytorch.org`

Table 1: Performance comparison on the SwitchBoard dataset (P: n-gram precision, R: n-gram recall, A: Average, E: Extrema, G: Greedy, L: average length)

| Model | BLEU | | | BOW Embedding | | | intra-dist | | inter-dist | | L |
|---|---|---|---|---|---|---|---|---|---|---|---|
| | R | P | F1 | A | E | G | dist-1 | dist-2 | dist-1 | dist-2 | |
| HRED | 0.262 | 0.262 | 0.262 | 0.820 | 0.537 | 0.832 | 0.813 | 0.452 | 0.081 | 0.045 | 12.1 |
| SeqGAN | 0.282 | **0.282** | 0.282 | 0.817 | 0.515 | 0.748 | 0.705 | 0.521 | 0.070 | 0.052 | **17.2** |
| CVAE | 0.295 | 0.258 | 0.275 | 0.836 | 0.572 | 0.846 | 0.803 | 0.415 | 0.112 | 0.102 | 12.4 |
| CVAE-BOW | 0.298 | 0.272 | 0.284 | 0.828 | 0.555 | 0.840 | 0.819 | 0.493 | 0.107 | 0.099 | 12.5 |
| CVAE-CO | 0.299 | 0.269 | 0.283 | 0.839 | 0.557 | 0.855 | 0.863 | 0.581 | 0.111 | 0.110 | 10.3 |
| VHRED | 0.253 | 0.231 | 0.242 | 0.810 | 0.531 | 0.844 | **0.881** | 0.522 | 0.110 | 0.092 | 8.74 |
| VHCR | 0.276 | 0.234 | 0.254 | 0.826 | 0.546 | 0.851 | 0.877 | 0.536 | 0.130 | 0.131 | 9.29 |
| DialogWAE | 0.394 | 0.254 | 0.309 | 0.897 | 0.627 | 0.887 | 0.713 | 0.651 | 0.245 | 0.413 | 15.5 |
| DialogWAE-GMP | **0.420** | 0.258 | **0.319** | **0.925** | **0.661** | **0.894** | 0.713 | **0.671** | **0.333** | **0.555** | 15.2 |

Table 2: Performance comparison on the DailyDialog dataset (P: n-gram precision, R: n-gram recall, A: Average, E: Extrema, G: Greedy, L: average response length)

| Model | BLEU | | | BOW Embedding | | | intra-dist | | inter-dist | | L |
|---|---|---|---|---|---|---|---|---|---|---|---|
| | R | P | F1 | A | E | G | dist-1 | dist-2 | dist-1 | dist-2 | |
| HRED | 0.232 | 0.232 | 0.232 | 0.915 | 0.511 | 0.798 | 0.935 | 0.969 | 0.093 | 0.097 | 10.1 |
| SeqGAN | 0.270 | 0.270 | 0.270 | 0.907 | 0.495 | 0.774 | 0.747 | 0.806 | 0.075 | 0.081 | 15.1 |
| CVAE | 0.265 | 0.222 | 0.242 | 0.923 | 0.543 | 0.811 | 0.938 | 0.973 | 0.177 | 0.222 | 10.0 |
| CVAE-BOW | 0.256 | 0.224 | 0.239 | 0.923 | 0.540 | 0.812 | **0.947** | **0.976** | 0.165 | 0.206 | 9.8 |
| CVAE-CO | 0.259 | 0.244 | 0.251 | 0.914 | 0.530 | 0.818 | 0.821 | 0.911 | 0.106 | 0.126 | 11.2 |
| VHRED | 0.271 | 0.260 | 0.265 | 0.892 | 0.507 | 0.786 | 0.633 | 0.771 | 0.071 | 0.089 | 12.7 |
| VHCR | 0.289 | 0.266 | 0.277 | 0.925 | 0.525 | 0.798 | 0.768 | 0.814 | 0.105 | 0.129 | 16.9 |
| DialogWAE | 0.341 | 0.278 | 0.306 | 0.948 | 0.578 | 0.846 | 0.830 | 0.940 | **0.327** | 0.583 | 18.5 |
| DialogWAE-GMP | **0.372** | **0.286** | **0.323** | **0.952** | **0.591** | **0.853** | 0.754 | 0.892 | 0.313 | **0.597** | **24.1** |

and $1 \times 10^{-5}$ for the generator and the discriminator, respectively. We tune the hyper-parameters on the validation set and measure the performance on the test set.

# 5 EXPERIMENTAL RESULTS

## 5.1 QUANTITATIVE ANALYSIS

Tables 1 and 2 show the performance of DialogWAE and baselines on the two datasets. DialogWAE outperforms the baselines in the majority of the experiments. In terms of BLEU scores, Dialog-WAE (with a Gaussian mixture prior network) generates more relevant responses, with the average recall of 42.0% and 37.2% on both of the datasets. These are significantly higher than those of the CVAE baselines (29.9% and 26.5%). We observe a similar trend to the BOW embedding metrics.

DialogWAE generates more diverse responses than the baselines do. The *inter-dist* scores are significantly higher than those of the baseline models. This indicates the sampled responses contain more distinct $n$-grams. DialogWAE does not show better intra-distinct scores. We conjecture that this is due to the relatively long responses generated by the DialogWAE as shown in the last columns of both tables. It is highly unlikely for there to be many repeated $n$-grams in a short response.

We further investigate the effects of the number of prior components ($K$). Figure 2 shows the performance of DialogWAE-GMP with respect to the number of prior components $K$. We vary $K$ from 1 to 9. As shown in the results, in most cases, the performance increases with $K$ and decreases once $K$ reaches a certain threshold, for example, three. The optimal $K$ on both of the datasets was around 3. We attribute this degradation to training difficulty of a mixture density network and the lack of appropriate regularization, which is left for future investigation.

## 5.2 QUALITATIVE ANALYSIS

Table 3 presents examples of responses generated by the models on the DailyDialog dataset. Due to the space limitation, we report the results of CVAE-CO and DialogWAE-GMP, which are the representative models among the baselines and the proposed models. For each context in the test set, we show three samples of generated responses from each model. As we expected, DialogWAE generates more coherent and diverse responses that cover multiple plausible aspects. Furthermore,

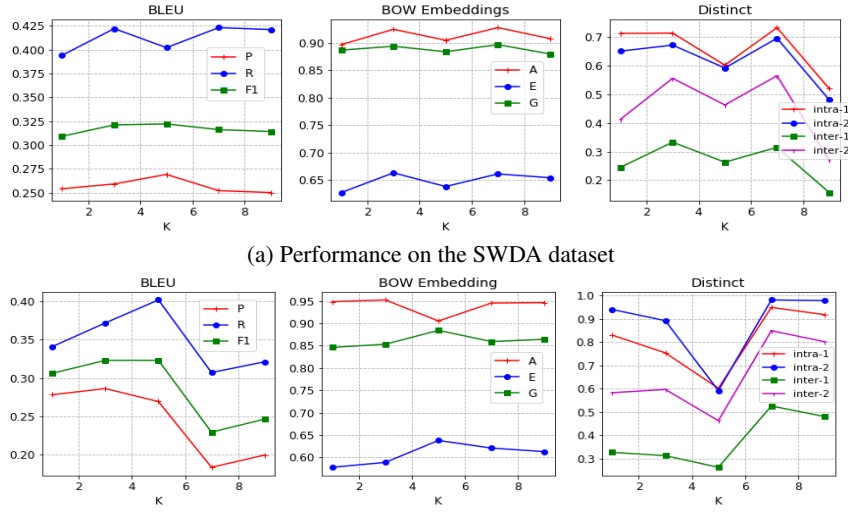

(a) Performance on the SWDA dataset

(b) Performance on the DailyDial dataset

Figure 2: Performance with respect to the number of prior components

Table 3: Examples of context-response pairs for the neural network models. _eou_ indicates a change of turn. 'Eg.i' means the $i$-th example.

| Context | Examples of Generated Responses | |
| --- | --- | --- |
| | CVAE-CO | DialogWAE-GMP |
| thank your for calling world airline. what can I do for you? _eou_ | Eg.1: i'm afraid i can't find it. Eg.2: what's the matter? Eg.3: hi, this is my first time. | Eg.1: i'd like to make a reservation for you, please Eg.2: do you know where i can get to get? Eg.3: can you tell me the way to the station? |
| how much is the rent? _eou_ the rent is $1500 per month. | Eg.1: how much is the rent? Eg.2: how much is the rent? Eg.3: what is the difference? | Eg.1: no problem. i'll take it. Eg.2: this one is $1.50.50,000 yuan per month. Eg.3: that sounds like a good idea. |
| guess who i saw just now ? _eou_ who? _eou_ john smith. _eou_ that bad egg who took the low road since he was a boy. | Eg.1: yes, he is. Eg.2: yes, he is Eg.3: yes, he is. | Eg.1: it is my favorite. Eg.2: no, but i didn't think he was able to get married. i had no idea to get her. Eg.3: this is not, but it's not that bad. it's just a little bit, but it's not too bad. |

we notice that the generated response is long and exhibits informative content. By contrast, the responses generated by the baseline model exhibit relatively limited variations. Although the responses show some variants in contents, most of them share a similar prefix such as *"how much"*.

We further investigate the interpretability of Gaussian components in the prior network, that is, what each Gaussian model has captured before generation. We pick a dialogue context "I'd like to invite you to dinner tonight, do you have time?" which is also used in (Shen *et al.*, 2018) for analysis and generate five responses for each Gaussian component. As shown in Table 4, different

Table 4: Examples of generated responses for each Gaussian component. 'Eg.i' means the $i$-th example.

| Context | I would like to invite you to dinner tonight, do you have time? | | |
| --- | --- | --- | --- |
| | Component 1 | Component 2 | Component 3 |
| Replies | Eg.1: Yes, I'd like to go with you. Eg.2: My pleasure. Eg.3: OK, thanks. Eg.4: I don't know what to do Eg.5: Sure. I'd like to go out | Eg.1: I'm not sure. Eg.2: I'm not sure. What's the problem? Eg.3: I'm sorry to hear that. What's the problem? Eg.4: It's very kind of you, too. Eg.5: I have no idea. You have to | Eg.1: Of course I'm not sure. What's the problem? Eg.2: No, I don't want to go. Eg.3: I want to go to bed, but I'm not sure. Eg.4: Of course not. you. Eg.5: Do you want to go? |

Gaussian models generate different types of responses: component 1 expresses a strong will, while component 2 expresses some uncertainty, and component 3 generates strong negative responses. The overlap between components is marginal (around 1/5). The results indicate that the Gaussian mixture prior network can successfully capture the multimodal distribution of the responses.

To validate the previous results, we further conduct a human evaluation with Amazon Mechanical Turk. We randomly selected 50 dialogues from the test set of DailyDialog. For each dialogue context, we generated 10 responses from each of the four models. Responses for each context were inspected by 5 participants who were asked to choose the model which performs the best in regarding to coherence, diversity and informative while being blind to the underlying algorithms. The average percentages that each model was selected as the best to a specific criterion are shown in Table 5.

Table 5: Human judgments for models trained on the Dailydialog dataset

| Model | Coherence | Diversity | Informative |
|---|---|---|---|
| CVAE-CO | 14.4% | 19.2% | 24.8% |
| VHCR | 26.8% | 22.4% | 20.4% |
| DialogWAE | 27.6% | **29.2%** | 25.6% |
| DialogWAE-GMP | **31.6%** | **29.2%** | **29.6%** |

The proposed approach clearly outperforms the current state of the art, CVAE-CO and VHCR, by a large margin in terms of all three metrics. This improvement is especially clear when the Gaussian mixture prior was used.

## 6 CONCLUSION

In this paper, we introduced a new approach, named DialogWAE, for dialogue modeling. Different from existing VAE models which impose a simple prior distribution over the latent variables, DialogWAE samples the prior and posterior samples of latent variables by transforming context-dependent Gaussian noise using neural networks, and minimizes the Wasserstein distance between the prior and posterior distributions. Furthermore, we enhance the model with a Gaussian mixture prior network to enrich the latent space. Experiments on two widely used datasets show that our model outperforms state-of-the-art VAE models and generates more coherent, informative and diverse responses.

### ACKNOWLEDGMENTS

This work was supported by the Creative Industrial Technology Development Program (10053249) funded by the Ministry of Trade, Industry and Energy (MOTIE, Korea).

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
