# OpenReview forum: "DialogWAE: Multimodal Response Generation with Conditional Wasserstein Auto-Encoder"
_ICLR.cc/2019/Conference_

### Official Review · AnonReviewer2 · 2018-11-02
**This paper presents a dialog response generation model based on the framework of adversarial autoencoder.**

**Rating:** 5
**Confidence:** 3

**Review:**

This paper presents a dialogue response generation model based on the framework of adversarial autoencoder. Specifically, the proposed model uses an autoencoder to encode and decode a response in a dialogue, conditioning on the context of the dialogue. The RNN encoded context is used as the prior of the latent variable in the autoencoder, and the whole dialogue (context + response) is used to infer the posterior of the latent variable. The inference is done by the adversarial training to match the prior and the posterior of the latent variable. Besides constructing the prior with a single Gaussian, the variant of the proposed model is also proposed where the prior is constructed with a Gaussian mixture model.

My comments are as follows:

1. The paper is well-written and easy to follow.

2. The experiments seem quite strong and the compared models are properly selected. I'm not an expert in the specific area of the dialogue generation. But to me, the results seem convincing to me.

3. The usage of the Wasserstein distance in the proposed model does not make sense to me. Both the adversarial training in AAE and minimising the Wasserstein distance are able to match the prior and posterior of the latent variable. If the former is used in the proposed model, then how is the Wasserstein distance used at the same time? I also checked Algorithm 1 and did not find how the Wasserstein distance comes in. This is the first question that needs the authors to clarify.

4. To me, the significance of this paper mainly goes to combining several existing frameworks and tricks into the specific area of dialogue generation. Although the empirical results show the proposed model outperforms several existing models, my concern is still on the originality of the paper. Specifically, one of the main contributions goes to using the Gaussian mixture to construct the prior, but this is not a whole new idea in VAE or GAN, nor using the Gumbel trick.

5. It is good to see that the authors showed some comparisons between DialogWAE and DialogWAE-GMP, letting us see GMP does help the performance. But a minor concern is that it seems hard to identify which part makes DialogWAE get superior performance than others. Are all the models running with the same experiment settings including the implementation of the RNNs?

---

> ### Author Response · Authors · 2018-11-09
> **Response to Reviewer 2**
>
> We thank the reviewer for taking the time to read our paper and for the useful comments to help improve our presentation! Below we address the specific points raised by the reviewer:
>
> >>>
>  3. The usage of the Wasserstein distance in the proposed model does not make sense to me…
> <<<
>
> We apologize for not making the algorithm clear. In our paper, we consider the Wasserstein Auto-Encoder (WAE) as a special AAE, as what WGAN is to GAN.  Therefore, training AAE (i.e., WAE in our algorithm) means minimizing the Wasserstein distance. They are not trained separately at the same time. We will clarify this more clearly in the upcoming revision.
>
> >>>
>  4. To me, the significance of this paper mainly goes to combining several existing frameworks and tricks into the specific area of dialogue generation…
> <<<
>
> First, we want to clarify that our core contribution is on the improved GAN architecture for a crucial problem in dialogue modeling rather than a trial-and-error methodology by combining different tricks.
>
> We agree that our paper's contribution is not geared toward machine learning itself but how it could be used better for dialogue generation, which is considered one of the core challenges in both machine learning, natural language processing, and more broadly artificial intelligence. As is clear from the call-for-papers which states ``applications'' as one of the major subject areas, we believe our work fits the conference's scope well.
>
> >>>
>  5. But a minor concern is that it seems hard to identify which part makes DialogWAE get superior performance than others…
> <<<
>
> Except for the GAN module, all the models are running with the same experimental setup including the implementation of RNNs. Therefore, the superior performance can be attributed to the improved GAN architecture.

---

### Official Review · AnonReviewer1 · 2018-11-02
**Clear ideas with convincing results**

**Rating:** 7
**Confidence:** 3

**Review:**

This paper proposes a novel dialogue modeling framework DialogWAE, which adopts conditional Wasserstein Auto-Encoder to learn continuous latent variables z that represents the high-level representation of responses. To enrich the diversity of the latent representations and capture multiple modes in the latent variables, the authors propose an advanced version (DialogWAE-GMP) of DialogWAE and models the prior distribution with a mixture of Gaussian distributions instead one.

Strength: The idea is clear and the paper is very well written. The authors evaluate the proposed models on a variety of reasonable metrics and compare against seven recently-proposed baselines.  Results show that both DialogWAE and DialogWAE-GMP generate responses that are both more similar to the references (BLEU and BOW embeddings) and more diverse (inter-dist). Human evaluations also show that the proposed models generate better responses than two representative baselines.

Minor comments/questions:

1) Missing citation, the optimization problem of this paper (Equation 5) is similar to the Adversarially Regularized Autoencoders (ICML 2018).

2) The authors use Gumbel-Softmax re-parametrization to sample an instance for the Gaussian Mixture prior network. Are you using the Straight-Through estimator or the original one? If the original Gumbel-Softmax estimator is used, it is better to show a comparison between simply using the Softmax with Gumbel softmax. Since the discrete sampling is not crucial in this case, a mixture of weighted representation may also work.

3) The DialogWAE-GMP with Gaussian Mixture prior network achieves great evaluation results and is better than the non-mixture version. I'd be interested to see some analysis on what each Gaussian model has captured. Will different Gaussian model generate different types of responses? Are the differences interpretable?

---

> ### Author Response · Authors · 2018-11-13
> **Response to Reviewer 1**
>
> We thank the reviewer for taking the time to read our paper and for the useful comments to help improve our presentation! Below we address the specific points raised by the reviewer:
>
> >>>
> 1) Missing citation, the optimization problem of this paper (Equation 5) is similar to the Adversarially Regularized Autoencoders (ICML 2018).
> <<<
>
> We have cited and discussed the ARAE paper in the revision.
>
> >>>
> 2) The authors use Gumbel-Softmax re-parametrization to sample an instance for the Gaussian Mixture prior network. Are you using the Straight-Through estimator or the original one?
> <<<
>
> We use the Strait-Through Gumbel softmax.
>
> >>>
> 3) I'd be interested to see some analysis on what each Gaussian model has captured. Will different Gaussian model generate different types of responses? Are the differences interpretable?
> <<<
>
> In the revised manuscript we have added an example showing responses for each Gaussian component. We select a dialogue context used in a baseline paper (Shen et al. 2018) for analysis and generate 5 responses for each component using DialogWAE-GMP. Results are shown in the following table (Reviewers can also find it from Table 4 in our revised manuscript):
> -----------------------------------------------------------------------------------------------------------------------------------------------------
>  Context  |        I would like to invite you to dinner tonight, do you have time?
> -------------|---------------------------------------------------------------------------------------------------------------------------------------
>                  |               Component 1              |                Component 2                     |         Component 3
>                  |-----------------------------------------|-----------------------------------------------|---------------------------------------------
>                  | Eg.1: Yes, I'd {like} to  go        | Eg.1: I'm {not sure}                          | Eg.1: {Of course} I'm {not} sure.
>                  |           with you.                          | Eg.2: I'm {not sure}.                         |           What's the problem?
>                  | Eg.2: My {pleasure}.                 |           What's the problem?             | Eg.2: {No}, I {don't} want to go
> Replies    | Eg.3: {OK}, thanks.                    | Eg.3: I'm sorry to hear that            | Eg.3: I want to {go to bed}, but
>                  | Eg.4: I don't know what to do|           What's the problem?             |           I'm not sure.
>                  | Eg.5: {Sure}. I'd {like} to          | Eg.4: It's very kind of you, too        | Eg.4: {Of course not}. you
>                  |           go out                               | Eg.5: I have {no idea}. You have to| Eg.5: Do you want to go?
> -----------------------------------------------------------------------------------------------------------------------------------------------------
>
> As shown in the table, different Gaussian models generate different types of responses: component 1 expresses a strong will, while component 2 expresses some uncertainty, and component 3 generates strong negative responses. The overlap between components is marginal (around 1/5). The results indicate that the Gaussian mixture prior network can successfully capture the multimodal distribution of the responses.

---

### Official Review · AnonReviewer3 · 2018-11-08
**A nice application of W-GAN to dialog, rather weak experiment analysis.**

**Rating:** 7
**Confidence:** 4

**Review:**

This paper uses Wasserstein GAN in conditional modeling of the dialog response generation. The main goal is to learn to use two network architecture to approximate the posterior distribution of the prior network. Instead of a KL divergence, like in VAE training, they use adversarial training and instead of using softmax output from the discriminator, they use Wasserstein distance. They also introduce a multi-modal distribution, GMM, while sampling from a the posterior during training, prior during the test time. The multi-modal sampling is based on gumbel-softmax over K possible G-distributions. They experiment on Daily Dialog and Switchborad datasets and show promising improvements on qualitative measures like BLEU and BOW embedding similarities, as well as qualitative measures including human evaluations comparing againsts substantial amount of baselines.

The paper presents a marriage of a few ideas together. First of, it uses the conditional structure presented in the ACL 2017 paper "Learning Discourse-level Diversity for Neural Dialog Models using Conditional Variational Autoencoders". It's great that they used that paper as their baseline. The extension is to use a GAN objective function (the discriminator) as critic and use Wasserstein GAN to to resolve the gradient vanishing issue and produce smooth gradients everywhere. In ACL 2017 paper they use KL divergence to make the posterior from the prior and rec-networks as close to each other so at test time the prior network can generate the samples similar to the true data features distribution. In this paper instead of KL, they use a Discriminator as in 'Adversarial AutoEncoders' paper. This paper extends AAE, instead uses the Wasserstein distance instead (1-Lipschitz function instead of softmax for the discriminator). The W-GAN has been shown to produce good results in text generation in this year's ICML 2018 with the paper 'Adversarially Regularized GAN' (AARE). The idea was to resolve VAE posterior collapse issue by using a discriminator as a regularizer instead of KL divergence with a stronger sampler from the output of the generator to map from noise sampler into the latent space. Interestingly, AARE paper is not cited in this work, which i think is an issue. I understand that paper was just for generation purpose not specific to the dialog modeling, but it makes the claims in the paper misleading such as: "Unlike VAE conversation models that impose a simple distribution over latent variables, DialogWAE models the data distribution by training a GAN within the latent variable space".

The part that i liked is the fact that they used multimodal gaussian distributions. I agree with the authors that using Gaussian for the approximating distribution only limits the sampling space and can weaken the models capability of variation. Although it is not proven for text, in image, the gaussian posteriors during training converge together into a single gaussian, causing blurry images. In this text this might correspond to dull responses in dialog. I would like the authors to comment on the interpretability of the components. Perhaps show a sample from each component (in the end the model decides which modal to choose before generation. Are these GMMs overlapping and how much ? Can you measure the difference between the means ?

I find the experiments extensive except the datasets are weaker.
I like the fact that they included human evaluations.

---

> ### Author Response · Authors · 2018-11-13
> **Response to Reviewer 3**
>
> We thank the reviewer for taking the time to read our paper and for the useful comments to help improve our presentation! Below we address the specific points raised by the reviewer:
>
> >>>
> Interestingly, AARE paper is not cited in this work, which I think is an issue…
> <<<
>
> We have cited and discussed the ARAE paper in the revision.
>
> >>>
> I would like the authors to comment on the interpretability of the components. Perhaps show a sample from each component (in the end the model decides which modal to choose before generation. Are these GMMs overlapping and how much? Can you measure the difference between the means?
> <<<
>
> In the revised manuscript we have added an example showing responses for each Gaussian component. We select a dialogue context used in a baseline paper (Shen et al. 2018) for analysis and generate 5 responses for each component using DialogWAE-GMP. Results are shown in the following table (Reviewers can also find it from Table 4 in our revised manuscript):
> -----------------------------------------------------------------------------------------------------------------------------------------------------
>  Context  |        I would like to invite you to dinner tonight, do you have time?
> -------------|---------------------------------------------------------------------------------------------------------------------------------------
>                  |               Component 1              |                Component 2                     |         Component 3
>                  |-----------------------------------------|-----------------------------------------------|---------------------------------------------
>                  | Eg.1: Yes, I'd {like} to  go        | Eg.1: I'm {not sure}                          | Eg.1: {Of course} I'm {not} sure.
>                  |           with you.                          | Eg.2: I'm {not sure}.                         |           What's the problem?
>                  | Eg.2: My {pleasure}.                 |           What's the problem?             | Eg.2: {No}, I {don't} want to go
> Replies    | Eg.3: {OK}, thanks.                    | Eg.3: I'm sorry to hear that            | Eg.3: I want to {go to bed}, but
>                  | Eg.4: I don't know what to do|           What's the problem?             |           I'm not sure.
>                  | Eg.5: {Sure}. I'd {like} to          | Eg.4: It's very kind of you, too        | Eg.4: {Of course not}. you
>                  |            go out                              | Eg.5: I have {no idea}. You have to | Eg.5: Do you want to go?
> -----------------------------------------------------------------------------------------------------------------------------------------------------
> As shown in the table, different Gaussian models generate different types of responses: component 1 expresses a strong will, while component 2 expresses some uncertainty, and component 3 generates strong negative responses. The overlap between components is marginal (around 1/5). The results indicate that the Gaussian mixture prior network can successfully capture the multimodal distribution of the responses.

---

### Author Response · Authors · 2018-12-17
**For All Reviewers**

We appreciate all reviewers for constructive feedback and comments for the improvement of the paper.
We have revised our paper according to the comments and replied to all reviewers.
Before the final decision, we will explain any of your questions.

---

### Public Comment · (anonymous) · 2018-12-21
**Confused with the evaluation metric - BLEU**

Thanks for the interesting paper and congratulations on the acceptance!

I have questions regarding your evaluation metric;

- What is Precision and Recall for BLEU? There is a description, but it's quite confusing. I'm assuming that precision must be the original BLEU, but not sure about Recall.
- Why use only up to n<4 for BLEU? In the original DailyDialog paper, I recall that BLEU-4 was near 0 for most baseline methods.

---

> ### Author Response · Authors · 2018-12-23
> **Explanation about BLEU**
>
> Thanks for the congratulation!
>
> Please refer to Section 5.2 in https://arxiv.org/pdf/1703.10960.pdf for the definition of precision and recall.
> We will revise the BLEU definition in the final version.
>
> We followed their implementation in Github which used bleu 1-3.
> We tested BLEU 1-4 and the results were similar.

---

### Meta-Review · Area_Chair1 · 2018-12-13
**a novel, improved GAN for dialogue modeling**

**Confidence:** 4
**Recommendation:** Accept (Poster)

**Metareview:**

This paper tackles the task of end-to-end systems for dialogue generation and proposes a novel, improved GAN for dialogue modeling, which adopts conditional Wasserstein Auto-Encoder to learn high-level representations of responses. In experiments, the proposed approach is compared to several state-of-the-art baselines on two dialog datasets, and improvements are shown both in terms of objective measures and human evaluation, making a strong support for the proposed approach.
Two reviewers suggest similarities with a recent ICML paper on ARAE and request including reference to it and also request examples demonstrating differences, which are included in the latest version of the paper.